# COVID-19 Vaccine Hesitancy: A Critical Time Period Analysis

**DOI:** 10.3390/ijerph19138098

**Published:** 2022-07-01

**Authors:** John R. Kues, Jacqueline M. Knapke, Shereen Elshaer, Angela M. Mendell, Laura Hildreth, Stephanie M. Schuckman, Julie Wijesooriya, Melinda Butsch Kovacic

**Affiliations:** 1Department of Family and Community Medicine, College of Medicine, University of Cincinnati, Cincinnati, OH 45229, USA; knapkeje@ucmail.uc.edu (J.M.K.); mendelam@ucmail.uc.edu (A.M.M.); hildrele@ucmail.uc.edu (L.H.); schuckse@ucmail.uc.edu (S.M.S.); 2Cincinnati Children’s Hospital Medical Center, Department of Pediatrics, College of Medicine, University of Cincinnati, Cincinnati, OH 45229, USA; elshaese@mail.uc.edu (S.E.); julie.wijesooriya@cchmc.org (J.W.); 3Department of Public Health and Preventive Medicine, Faculty of Medicine, Mansoura University, Mansoura City 35516, Egypt; 4Department of Rehabilitation, Exercise, and Nutrition Sciences, College of Allied Health Sciences, University of Cincinnati, Cincinnati, OH 45267, USA

**Keywords:** vaccine hesitancy, COVID-19, pandemic, cross-sectional survey design, community-engaged

## Abstract

The COVID-19 pandemic has been a devastating, global public health crisis. Public health systems in the United States heavily focused on getting people to adhere to preventive behaviors, and later, to get vaccinated. January through May of 2021 was a critical and volatile time period for COVID-19 cases, deaths, and expanding vaccination programs coinciding with important political and social events which will have a lasting impact on how the public views science, places trust in our government, and views individual rights. Having collected almost 1400 surveys, our goal was to assess vaccine behavior, explore attitudes toward receiving the vaccine, and identify trusted information sources. More than 83% of our survey respondents said they were at least partially vaccinated. Of 246 unvaccinated, 31.3% were somewhat or extremely likely to get vaccinated when available. Their two most common concerns were vaccine effectiveness (41.1%) and safety (40.2%). Significant differences were observed between respondents who were likely to be vaccinated in the future and those who were hesitant on three of five demographic variables. Our data provide unique insight into the history of behavior and motivations related to COVID-19 vaccines—what will be seen as a “wicked problem” for years to come.

## 1. Introduction

The COVID-19 pandemic in the United States (US) has been the most devastating public health problem in over a century. By the end of 2021, there were more than 800,000 US deaths [1]. The impact of the pandemic has been far-reaching, causing numerous crises in healthcare delivery. People needing surgical and medical treatment as well as other hospital services could no longer access them. Beyond healthcare, COVID-19 impacted key aspects of the economy, including unemployment, the entertainment and food service industries, and transportation [2]. Public policies regarding precautions and vaccinations varied widely by state and city with people taking sides based on political affiliations. Indeed, the COVID-19 pandemic was a major issue in the 2020 US presidential election, and political affiliation was observed to be a major predictor of trust in various information sources related to the pandemic and adherence to risk mitigation recommendations [3].

Much attention was given to the importance of people getting vaccinated and adhering to preventive behaviors. Primarily voluntary measures, most were subject to the influence of social, political, and economic environments. Research conducted to address the COVID-19 pandemic frequently surveyed individuals’ behavior, attitudes, and beliefs related to the virus [4,5,6]. Some sought to better understand psychological factors such as conspiracy and apocalyptic beliefs, which have each been shown to influence individuals’ behaviors and attitudes [7]. Rarely have reports of these studies been examined in the larger context of other environmental variables. One 15-study literature review found that demographics, employment status, political views, religiosity, and education/income levels were all associated with COVID-19 vaccine uptake [8]. Another review also found demographics to be correlated with vaccine hesitancy as well as personal judgments regarding trustworthiness of authorities, aversion to risk, and disgust sensitivity [9]. Others have reported that these dynamic environmental variables have made the development and implementation of public health interventions challenging and their impact inconsistent [10].

From the initial spread of COVID-19 in early 2020 through the latter part of that year, when hospitalizations and deaths were rapidly rising, and into 2021 as scientists and healthcare workers feverishly worked to develop, test, and distribute various COVID-19 vaccines, there were many critical periods where in-depth analysis of the epidemiology of the virus overlapped with social and political events. These overlaps have confounded our ability to learn from, and effectively battle, the pandemic. Our study focused on one of the most critical and volatile periods: January through May of 2021.

Epidemiologically, this period saw the most dramatic drop in COVID-19 cases and deaths [1]. During that time, the daily average of newly reported cases of COVID-19 also reached almost 300,000 in early January 2021. Those numbers dropped significantly and leveled off at 65,000–70,000 per day by March. Importantly, there were over 3600 daily COVID-19 deaths in early January 2021, which later dropped to under 1000 by mid-March and to under 500 by early May [1]. Additionally, in early January, there was limited availability of COVID-19 vaccines and distribution challenges, resulting in a slow initial start to public vaccination programs. Nevertheless, almost 500,000 doses of COVID-19 vaccines were still being distributed daily during that time. By early April, the number had risen considerably—to over 1.8 million daily doses. By mid-May, however, the daily doses fell to approximately 600,000 per day [1]. While the number of fully vaccinated individuals in the US was only 52,000 in early January, those numbers grew to over 145.9 million by the end of May 2021, when vaccine distribution widened [1]. Vaccine mandates for healthcare workers, and others, created a sharp divide between vaccine supporters and those who were hesitant or completely resistant. 

During this critical time of dramatic swings in COVID-19 cases, deaths, and vaccinations, important political and social issues had already politicized public health measures, information sources, and the COVID-19 vaccine. The political campaigns in 2020 contributed to polarized opinions regarding the severity of the COVID-19 pandemic, the effectiveness of mitigation measures such as masking and social distancing, and the safety of COVID-19 vaccines. Misinformation and conspiracy theories related to the 2020 presidential election resulted in the January 6th incident at the US Capitol and recorded high levels of distrust in the government. An increasing portion of the population turned to websites and social media for information [11,12]. As the rate of vaccinations began to decline in April 2021, a combination of mandates and incentives were initiated by private employers and governmental entities. Neither of these had a significant impact on vaccinations. The gap between those who were vaccinated, or intended to be, and those who were hesitant or strongly opposed was becoming wider and stronger by mid-Spring. Additionally, many local and state governments were taking measures to ban mask mandates and other COVID-related mitigation measures. This was also a period when there were national and international discussions about vaccine passports that would limit travel, participation in some social and/or recreational activities, and access to some venues and locations. While vaccine passports never came to fruition, it angered many who were resisting vaccinations and some who were already vaccinated [13]. 

Examination of events during this important period is critical to our understanding of people’s attitudes and behaviors. These events have had a lasting impact on how the public views science, trusts our government, and views individual rights. It has transformed public health initiatives into political statements, and it has limited many people’s ability and interest in becoming educated about serious health issues. Discourse about COVID-19 was often avoided.

During April and May of this critical period, we gathered surveys from almost 1400 respondents. The goal of our survey was to assess vaccine behavior at that time as well as explore peoples’ attitudes toward receiving the vaccine. Just as important, we wanted to identify peoples’ trusted information sources. We found that the data we collected provided useful insights into our understanding of the history of behavior and motivations related to COVID-19 vaccines and has helped us better understand the relationships between these variables and the demographic characteristics of our survey population.

## 2. Materials and Methods

An 11-item questionnaire was developed to collect data for this study. It included five demographic questions (gender, race/ethnicity, age, education, and home zip code). We asked a single question (with multiple responses possible) about participants’ experiences with COVID-19 testing, diagnosis, and health consequences for themselves and friends/family. We asked participants whether they had received a full (a single vaccination of the Johnson and Johnson vaccine, or 2 vaccinations of the Moderna or Pfizer vaccines) or partial vaccination (only a single vaccination of the Moderna or Pfizer vaccines), and if they were not vaccinated, we asked them the likelihood of getting the vaccine in the future. For those who were currently unvaccinated, we provided a list of 15 potential reasons that would make them less likely to get a COVID-19 vaccine (including “other; specify”). We provided all respondents (vaccinated and unvaccinated) with a list of eight reasons that motivated them to get the vaccine (if they were already vaccinated) or more likely to get the vaccine (if they were currently unvaccinated). The responses included “other; specify.” Finally, we provided participants with a list of 14 information sources and asked them to rate their level of trust in information regarding COVID-19 and vaccines. This list included local and national public health sources (Centers for Disease Control (CDC), national public health experts, and public health officials), healthcare experts (pharmacists, personal/family healthcare providers, friends/family with medical/science training), and social/governmental sources (government officials, social media, pastors/ministers, employers, news media, etc.). To ensure local community relevance, members of a local community advisory board advised the selection of questions and response choices and pre-tested the survey for ease of use prior to its release.

The survey was configured to be delivered online (via Research Electronic Data Capture software, or REDCap™ 11.1.14 hosted by Cincinnati Children’s Hospital Medical Center, Cincinnati, OH, USA) and on paper. A link to the online survey was posted on websites, listservs, social media sites, electronic newsletters, and other electronic platforms maintained by community organizations and was available for completion from 1 April 2021 through 31 May 2021. With a specific intention to increase minority participation in the survey, five trained local community advisory board members administered paper surveys in Cincinnati.

Only responses from adults over the age of 18 years old and living in the US were included in the analysis. The data were analyzed using SPSS^®^ 26.0 (IBM, Armonk, NY, USA). Bivariate analyses for categorical and ordinal variables were calculated using chi-square with *p*-values of <0.05 for statistical significance.

## 3. Results

We received 1150 surveys from online sites and 249 paper surveys from community liaisons. Partially completed surveys were included in our analysis when possible. We received survey responses from 43 states plus Washington, DC; however, the largest portion of responses (42.0%) was from the Greater Cincinnati Area. Table 1 describes the demographics of the 1399 survey respondents, who were mostly female, with the largest percentage being White. The respondents tended to be younger (more than 70% were 45 or younger). The majority had less than a baccalaureate degree. The largest number of respondents were from Ohio (541), with 471 responses from Hamilton County (Cincinnati). 

Respondents indicated their past experience with COVID-19. Over half (742, or 53%) had been tested at some point and 211 (15.1%) had been diagnosed with COVID-19. Other experiences included someone else in the home who had been diagnosed with the virus (268, or 19.2%), having someone in the household hospitalized with COVID-19 (123, or 8.8%), and having someone close to them die from COVID-19-related complications (182, or 13%). However, almost one-fifth of the respondents had none of those experiences (274, or 19.6%).

### 3.1. Reasons for and against Getting Vaccinated

We asked all respondents what reasons they had for getting vaccinated or reasons they might decide to get vaccinated if they were currently unvaccinated. There were 3695 responses among the 1399 respondents. Over half of the respondents (52.3%) indicated that it would reduce their chances of contracting COVID-19. There were a number of other responses related to personal and community safety as well as comfort in returning to public activities. Finally, only a relatively small percentage (18.5%) indicated that it was required for employment (Table 2).

Currently unvaccinated respondents (*n* = 246) were asked to indicate reasons why they were hesitant or would not be vaccinated. There was a total of 802 responses to this question. Many responses reflected a distrust or concern about vaccines in general, and 11.9% of the unvaccinated respondents did not believe that COVID-19 was dangerous to them. The two most common concerns were that respondents were unsure about the effectiveness (41.1%) and safety (40.2%) of the vaccines (Table 3). 

Finally, we asked respondents to indicate their trust in specific individuals and sources for truthful information about COVID-19 and vaccines. Of the 14 sources listed, only 3 (healthcare providers, the CDC, and national experts such as Dr. Fauci) were highly trusted by more than half of the respondents (Table 4). Social media and pastors/ministers had the highest percentage of respondents rating them as low trust (37.7% and 31.6%, respectively). Other local sources, such as friends and family with medical training, pharmacists, and local public health officials, were also perceived as more trustworthy regarding COVID-19 and vaccine information (Table 4).

### 3.2. Examination of Currently Unvaccinated Respondents

Of the 1399 survey respondents, 246 (17.6%) indicated that they were unvaccinated. Of those, about one-third (77/246) said that they were somewhat or extremely likely to get the vaccine when it was available to them. Ninety (36.6%) said that they were somewhat or extremely unlikely to get the vaccine, and 56 (22.8%) were neutral on their vaccine intentions. For the purposes of analysis, we combined the “somewhat” and “extremely” responses together. We also included neutral responses with the “unlikely” responses in order to compare hesitant and non-hesitant individuals. 

We found significant differences between respondents who were likely to get the vaccine in the future and those who were hesitant on three of five demographic variables (Table 5). We found that unvaccinated men were more likely than unvaccinated women to indicate that they were likely to get the vaccine in the future (42.9% vs. 27.8%). We found that White and other respondents were more likely to be vaccinated than Black respondents (46.4% and 51.4% vs. 23.2%). We also found that Hispanic respondents were more likely to indicate they will get vaccinated than non-Hispanic respondents (55.0% vs. 32.5%).

We compared respondents who were vaccinated, likely to get vaccinated when it is available to them, and those who were hesitant about their experiences with COVID-19. The only statistically significant difference among the three groups was for those who had no firsthand experience with the virus: we found that a smaller percentage of vaccinated (16.8%) and likely to get vaccinated (18.2%) had no experience with COVID-19 compared with over a third (33.6%) of those who were hesitant to get vaccinated (*p* < 0.001).

## 4. Discussion

Our study results show that the respondents of our survey had a very high vaccination rate (83.4%) at a time when the national vaccination rate was slightly over 60%. However, the 12% of our sample who indicated that they were unlikely to get vaccinated is consistent with other surveys for that time period. In fact, most surveys conducted in the US since early 2020 have found a very stable portion of the population who have steadfastly refused to get vaccinated (10–15%). As national vaccination rates continue to slowly increase (approximately 65% as of mid-February 2022), the percentage of the population who seem willing to consider receiving COVID-19 vaccinations continues to shrink relative to the stable population unwilling to get vaccinated. 

Public health problems such as the COVID-19 pandemic are complex, “wicked” problems, and public health officials, particularly in the US, need to study the context in which community members receive and interpret scientific information. Research has demonstrated that many factors impact an individual’s decision on whether to obtain a vaccine or not, and this study provides further evidence that those decisions are not based on just one variable. In terms of understanding how to handle public health issues in the future, we have learned we have to understand the context at the time. The COVID-19 pandemic has become one of the latest examples of a complex, or “wicked,” problem. So-called “wicked problems” are characterized by their complexity, non-linearity, and the difficulty in assessing the full impact of interventions [14]. Societies have a great deal of difficulty addressing wicked problems because governments are not structured to fully analyze such problems, and the strategies for developing and delivering solutions are limited and are not adaptable to the nature of these problems [15,16].

There has never been a public health crisis that has been complicated by as many social, political, and economic factors as COVID-19. In an era of instant news cycles, social media, and unprecedented political discord there is a low level of trust in traditional authoritative sources and suspicion of the medical research establishment. The current level of chaos and confusion is likely to be a common feature of these types of challenges in the future. It is therefore critical that we carefully examine the current crisis so that we are better prepared to address these types of complex health and social problems in the future.

Complex and wicked problems cannot be understood from a single perspective. It is the interplay and overlap of multiple contributing factors that create problems that are impossible to solve and difficult to address. The COVID-19 pandemic has been very well documented from healthcare and epidemiological perspectives. The CDC and the World Health Organization (WHO), as well as several other healthcare organizations, have been monitoring testing, cases, hospitalizations, and deaths since late 2019. Additionally, most of the political and social events during this time period have been reasonably well documented, although their interpretation and analysis have been complicated by the perspective of the news media and other sources of reporting. The assessment of the attitudes and experiences of the general US population has not been as well documented. There have been several studies conducted on segments of the US population at various times during the pandemic, but survey participants vary, as do the questions and focus of the surveys. Additionally, there are some significant gaps in the timeline that is covered by these studies. This leaves us with a somewhat spotty and disorganized picture of how people were reacting to the epidemiological, social, political, and economic components of the pandemic. These studies also lack a contextual framework of the timeframe from which they gathered their data. It is therefore difficult to fully interpret the findings. 

## 5. Conclusions

This study provides key insights into the behaviors and perceptions of the US population during a critical time of the COVID-19 pandemic, just as vaccines were becoming more available. Still, the survey was cross-sectional by design and reflects the perspectives of those that willingly participated, which could be biased given that most respondents participated online and were from the US only. Continued reflection after examining studies completed across the timeframe of the pandemic will be useful in addressing similar future wicked public health problems.

## Figures and Tables

**Table 1 ijerph-19-08098-t001:** Respondent demographics (*N* = 1399).

Variable	*n*	%
Gender		
Female	819	58.6
Male	557	39.9
Preferred gender not listed	21	1.5
Race		
White	646	46.3
Black	344	24.7
American Indian/Alaska Native	177	12.7
Asian	87	6.2
More than one race	87	6.2
Native Hawaiian/Pacific Islander	38	2.7
Preferred identity not listed	15	1.1
Age (years)		
18–30	509	36.4
31–45	484	34.6
46–60	206	14.7
61 and older	198	14.2
Education		
Less than a Bachelor’s Degree	794	57.0
Bachelor’s Degree	346	24.8
Post Bachelor’s Degree	254	18.2

**Table 2 ijerph-19-08098-t002:** Reasons given to get vaccinated.

Statements (*N* = 1399)	*n*	%
It will reduce my chances of getting COVID-19.	731	52.3
It will help end the pandemic sooner.	604	43.2
I will be less likely to transmit COVID-19 to others.	596	42.6
It will reduce my chances of becoming seriously ill from COVID-19 if I did.	587	42.0
I will be more comfortable going to public places.	482	34.5
It can help me be able to see my friends and family.	436	31.2
It is required by my employer.	259	18.5

**Table 3 ijerph-19-08098-t003:** Reasons for vaccine hesitancy among unvaccinated respondents.

Statements (*N* = 246)	*n*	%
I am unsure about the effectiveness of the COVID-19 vaccines.	101	41.1
I am unsure of the safety of the COVID-19 vaccines.	99	40.2
I am worried about the side effects of the vaccine.	92	37.4
I don’t know enough about the COVID-19 vaccines.	69	28.0
I worry there might be long-term effects from the COVID-19 vaccines.	66	26.8
I know someone who had a bad reaction to a COVID-19 vaccine.	55	22.4
I distrust vaccines because of the history of research on minority communities.	52	21.1
I don’t believe in the effectiveness of vaccines in general.	50	20.3
I have, or someone I know has had a bad reaction to other vaccines.	40	16.3
I have a condition that will keep me from being eligible (e.g., allergic reaction, chemotherapy, etc.).	39	15.9
It’s too difficult or confusing to get an appointment to get the COVID-19 vaccine.	39	15.9
I worry about the availability of the second vaccine shot after I get the first one.	36	14.6
The genetic (mRNA) component of some vaccines makes me uncomfortable.	35	14.2
I don’t believe that COVID-19 is dangerous to me.	29	11.8

**Table 4 ijerph-19-08098-t004:** Trust in information sources for COVID-19 and COVID-19 vaccines.

Information Source	Low Trust		I Don’t Know		High Trust	
	*n*	%	*n*	%	*n*	%
Social media	527	37.7	526	37.7	344	24.6
Pastor/Minister	441	31.6	606	43.4	349	25.0
News media	375	26.9	620	44.4	401	28.7
Schools	353	25.3	612	43.9	429	30.8
Leaders in my community	351	25.2	623	44.7	419	30.1
Employer	341	24.5	657	47.1	396	28.4
Friends and family	307	22.0	599	42.9	491	35.1
Center for Disease Control (CDC)	281	20.1	338	24.2	776	55.6
State government officials	276	19.8	552	39.5	569	40.7
Local public health officials	231	16.5	550	39.4	615	44.1
Friends and family with medical/science knowledge	201	14.4	515	36.9	681	48.7
National experts (Dr. Fauci)	183	13.1	440	31.5	772	55.3
Pharmacists	180	12.9	552	39.5	664	47.6
Healthcare providers	133	9.5	440	31.5	824	59.0

**Table 5 ijerph-19-08098-t005:** Hesitancy and demographic characteristics for unvaccinated respondents.

Variables (*N* = 246)	Unlikely/Neutral		Likely		χ^2^	*p*-Value
	*n*	%	*n*	%		
Age (years)					2.3	0.3
18–30	47	62.70%	28	37.30%		
31–45	59	62.80%	35	37.20%		
46 and older	40	74.10%	14	25.90%		
Gender					5.3	**0.02**
Female	83	72.20%	32	27.80%		
Male	56	57.10%	42	42.90%		
Race/Ancestry					14.9	**0.001**
White	37	53.60%	32	46.40%		
Black	86	76.80%	26	23.20%		
Other *	17	48.60%	18	51.40%		
Ethnicity					4.1	**0.04**
Hispanic/Latino	9	45.00%	11	55.00%		
Non-Hispanic/Latino	137	67.50%	66	32.50%		
Education					0.1	0.9
Less than a Bachelor’s degree	106	65.80%	55	34.20%		
Bachelor’s Degree	28	63.60%	16	36.40%		
Post Bachelor’s Degree	11	64.70%	6	35.30%		

Chi-square test was used, *p* < 0.05 (in bold) was considered statistically significant. * Other includes Asian, American Indian/Alaska Native, Native Hawaiian/Pacific Islander, and Multiracial.

## Data Availability

The datasets generated and analyzed during the study are available from the corresponding authors on reasonable request.

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
