# Peer review of "COVID-19 Vaccine Hesitancy: A Critical Time Period Analysis"

_ijerph, 2022, doi:10.3390/ijerph19138098_

Round 1

Reviewer 1 Report

Kues et al. reported that COVID-19 vaccine hesitancy.

 1.       In abstract section, please make a change from ”in over a century” to “recently”.

2.       In M&M section, “We asked participants whether they had received a full or partial vaccination, and if they were not vaccinated…” Please define “full vaccination and “partial vaccination.”

Author Response

Thank you for the useful comments. The following revisions have been made:

 “The COVID-19 pandemic has been the most devastating public health problem in over a century” was revised to “The COVID-19 pandemic has been a devastating, global public health crisis.”

The M&M was revised to read “We asked participants whether they had received a full (a single vaccination of the Johnson and Johnson vaccine, or 2 vaccinations of the Moderna or Pfizer vaccines) or partial vaccination (only a single vaccination of the Moderna or Pfizer vaccines), and if they were not vaccinated,…”

Reviewer 2 Report

The paper is interensting, welfare written and pleasant to read. describes a reality that concerns the United States but, with the due differences, I think it fits well in any place. I have no suggestions or comments. the paper in my opinion is interesting and well written. I also believe that the conclusions drawn by the Authors are valid and reflect the reality that we have all observed in these two years of pandemic. The more important issue may be that it is hat the analysis was conducted on an American population, and some information or items entered are not generalizable (especially the aspect of public health management).
However, public administration's handling of the pandemic has been largely shared around the world, and media information has influenced the general population everywhere.
For these reasons I recommend perhaps including these limitations in the discussion.

Author Response

Thank you for the useful comments. We better emphasized that this study was based on a US population in the following revisions:

We added “in the United States” to the second sentence of the abstract which now reads “Public health systems in the United States heavily focused on getting people to adhere to preventive behaviors…”

A single sentence in the last paragraph of the Methods section was added “Only responses from adults over the age of 18 years old and living in the US were included in the analysis.”

We added “in the US” to the third sentence of the discussion which now reads “In fact, most surveys conducted in the US since early 2020 have found a very stable portion of the population who have steadfastly refused to get vaccinated (10%-15%).”

We added “US” to describe the population in the following: “The assessment of the attitudes and experiences of the general US population has not been as well documented.”

We added “particularly in the US” to the first sentence of the second paragraph of the discussion which now reads “Public health problems like the COVID-19 pandemic are complex, wicked problems, and public health officials, particularly in the US, need to study the context in which community members receive and interpret scientific information.”

The first sentence of the conclusion was revised from “This study provides key insights into peoples’ behaviors and perceptions during a critical time of the COVID-19 pandemic, just as vaccines were becoming more available.” To “This study provides key insights into the behaviors and perceptions of the US population during a critical time of the COVID-19 pandemic, just as vaccines were becoming more available.” Further, “and were from the US only” was added to the second sentence in the conclusion.

Reviewer 3 Report

Dear authors,

in this cross-sectional survey you analyzed the vaccination behavior of over 1000 people. The age category of the participants is explained in the table. My advice is to declare, in the text, the criterion to include only adults over 18 years of age. Best regards

Author Response

Thank you for the useful comment. The following revision has been made:

As indicated above, a single sentence in the last paragraph of the Methods section was added “Only responses from adults over the age of 18 years old and living in the US were included in the analysis.”